# Strong charge carrier scattering at grain boundaries of PbTe caused by the collapse of metavalent bonding

Riga Wu[1,4], Yuan Yu [1,4] ✉, Shuo Jia[1], Chongjian Zhou [2], Oana Cojocaru-Mirédin [1] ✉ & Matthias Wuttig [1,3] ✉

Grain boundaries (GBs) play a significant role in controlling the transport of mass, heat and charge. To unravel the mechanisms underpinning the charge carrier scattering at GBs, correlative microscopy combined with local transport measurements is realized. For the PbTe material, the strength of carrier scattering at GBs depends on its misorientation angle. A concomitant change in the barrier height is observed, significantly increasing from low- to high-angle GBs. Atom probe tomography measurements reveal a disruption of metavalent bonding (MVB) at the dislocation cores of low-angle GBs, as evidenced by the abrupt change in bond-rupture behavior. In contrast, MVB is completely destroyed at high-angle GBs, presumably due to the increased Peierls distortion. The collapse of MVB is accompanied by a breakdown of the dielectric screening, which explains the enlarged GB barrier height. These findings correlate charge carrier scattering with bonding locally, promising new avenues for the design of advanced functional materials.

Grain boundaries (GBs), i.e., the interfaces between misoriented but otherwise identical crystals, are ubiquitous planar defects. They play a decisive role in determining the mechanical[1], magnetic[2], and electrical properties[3] of polycrystalline materials. The study of GBs and their impact on the properties of materials is hence an important and burgeoning topic in materials science. In thermoelectrics, GBs are frequently used to tailor the transport of electrons and phonons[4–6]. This facilitates modifying the thermoelectric figure-of-merit (ZT), which characterizes the ability of a material to convert heat into electrical energy[7]. Yet, studies of GB effects on the properties usually refer to the average grain size as the control parameter[8,9]. For example, the classical Hall-Petch relationship describes the strengthened yield stress of materials by increasing the density of GBs to impede the propagation of dislocations[10]. However, GBs are classified by five macroscopic degrees of freedom and they often show complexity in atomic structure, composition, and GB complexion transitions[11–13]. These factors should play at least an equally

important role in controlling GB-related properties compared with the grain size effect.

With the emergence of advanced characterization techniques, such as aberration-corrected (scanning) transmission electron microscopy (S/TEM) and atom probe tomography (APT), structural features and compositions of GBs can be characterized on an atomic scale[14,15]. This suite of techniques thus enables unravel intricate relationships between structure–property and composition. We will exemplify this for PbTe, where its performance is determined by the dimensionless thermoelectric figure-of-merit, defined as $ZT = S^2\sigma T/\kappa$, where S is the Seebeck coefficient, σ the electrical conductivity, T the absolute temperature, and κ the thermal conductivity[16]. These parameters are strongly interrelated and often show opposite trends[17]. A large σ, for example, is usually accompanied by a large κ of materials. Certain GB-related microstructures such as dislocation arrays formed at low-angle GBs (LAGBs)[18], twin boundaries[19], and segregation-induced GB complexions[20] have been suggested to lower the thermal conductivity

[1]Institute of Physics (IA), RWTH Aachen University, Sommerfeldstraße 14, 52074 Aachen, Germany. [2]State Key Laboratory of Solidification Processing, and Key Laboratory of Radiation Detection Materials and Devices, Ministry of Industry and Information Technology, Northwestern Polytechnical University, Xi'an 710072, China. [3]Peter Grünberg Institute (PGI 10), Forschungszentrum Jülich, 52428 Jülich, Germany. [4]These authors contributed equally: Riga Wu, Yuan Yu. ✉e-mail: yu@physik.rwth-aachen.de; cojocaru-miredin@physik.rwth-aachen.de; wuttig@physik.rwth-aachen.de

and enhance the ZT of materials. In contrast, in materials such as $Mg_3Sb_2$, the introduction of GBs has little impact on reducing the thermal conductivity but strongly increases the electrical resistivity and thus degrades σ and the ZT value[21]. Insights into the GB chemistry as revealed by APT indicate that the Mg deficiency at GBs leads to enhanced GB resistance[22]. While the structural and chemical information of a specific GB can be characterized by correlative TEM-APT methods[23], the electrical transport properties were often measured for the entire polycrystalline samples containing complex microstructures. Such samples, which include many different types of GBs and other defects (such as pores and precipitates) impede the desired 1:1 correlation of structure−property−composition for GBs. Characterizing the GB microstructure, composition, and transport properties at the same individual GB is thus a prerequisite to understand the impact of a specific GB on charge carrier transport[24].

## Results

### Correlative microscopy and characterization

To achieve this goal, we designed a correlative characterization method to determine local microstructures and transport properties of the same GB, as illustrated in Fig. 1. The potential of this approach is exemplified by Ag-doped PbTe, a typical thermoelectric material used for power generation[25]. Electron backscattered diffraction (EBSD) will be utilized to characterize the orientation of different grains and the classification of GBs, e.g., LAGBs and high-angle GBs (HAGBs) (Fig. 1a). This allows us to rapidly identify and characterize a GB, a prerequisite for the study of the impact of well-characterized GBs on charge transport. In Fig. 1b, a random HAGB with a misorientation angle of 45° is shown. This selected GB is "lifted-out" by a micromanipulator in a dual-beam focused ion beam (FIB) system, Fig. 1c. A well-shaped cuboid lamella with defined dimensions can be prepared by carefully cleaning the surfaces using FIB. This lamella is then transferred onto a microchip with pre-deposited gold electrodes by electron-beam lithography (EBL, details can be found in the Methods). Figure 1d presents the final configuration of the measurement device in a Hall-bar shape. The position of the GB is located in the middle of two voltage electrodes as indicated by an arrow. Within this configuration, we can measure the temperature-dependent (10−300 K) longitudinal electrical resistivity, Hall carrier concentration, and mobility by applying a constant current and swapping magnetic fields in a physical property measurement system (PPMS). The temperature range can be extended by employing setups for high-temperature transport property measurements. The same preparation and measurement procedure is applied to a single-grain sample to extract the contribution of the GB to the transport properties. To correlate the transport properties with local GB microstructures, a needle-shaped APT specimen is prepared by a standard site-specific "lift-out" method[26] from the same GB used for PPMS measurement (Fig. 1e). The GB misorientation and position of the APT specimen are further confirmed by transmission Kikuchi diffraction (TKD), as illustrated in Fig. 1f. The procedure outlined here can be easily adapted to many other solids beyond thermoelectrics.

### Misorientation angle dependent charge carrier transport

This correlative approach can be used to decipher the impact of individual, well-characterized GBs on transport properties. Figure 2a shows the temperature-dependent Hall carrier mobility for GBs with varying misorientation angles (θ). One single-grain sample is also measured for reference. The carrier mobility monotonously decreases with increasing temperature for the single-grain sample, due to the increased thermal occupation of phonons leading to an increase in electron-phonon scattering[27]. While the values of carrier mobility for samples with GB misorientation angles of 3.5° and 4.8° are slightly lower than that of the single-grain sample, the temperature dependence of their mobility is similar. This implies that the dominant scattering mechanism in these samples is the same, i.e., scattering of charge carriers dominated by acoustic phonons. In stark contrast, the three samples with HAGBs (θ > 15°) show a thermally activated carrier mobility below about 100 K. This temperature-activated carrier mobility is often induced by the scattering of charged defects in the

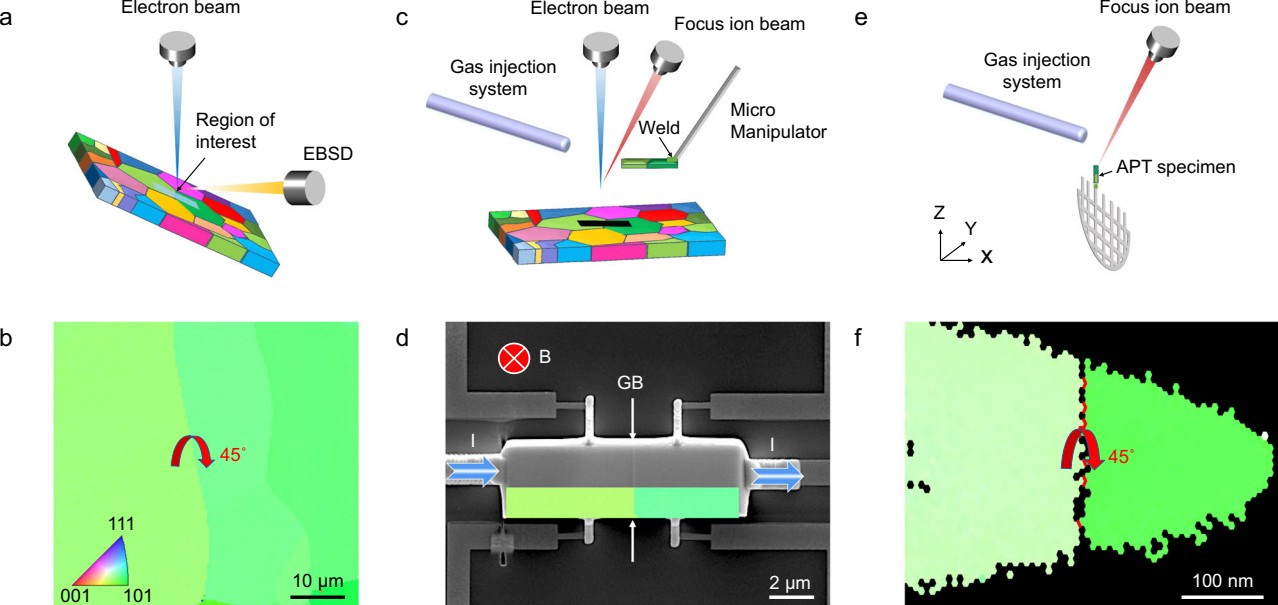

**Fig. 1 | Schematic diagrams illustrating the fabrication process of individual GB devices. a** Selection of the target GB with structural characteristics identified using SEM, EBSD, and FIB. **b** EBSD inverse pole figure (IPF) map indicating a GB with a misorientation angle of 45°. **c** The "lift-out" and transfer process of individual GB lamella with the help of FIB and gas injection system. **d** SEM image of a Hall-bar geometry device with a lamellar sample containing a 45° GB mounted on a Si/SiO₂ substrate pre-deposited with gold electrodes. **e** Corresponding APT specimen cut from the same GB used for PPMS measurement. **f** TKD image of the APT specimen showing the misorientation angle and position of the GB, confirming the same feature for both samples used for PPMS and APT measurement.

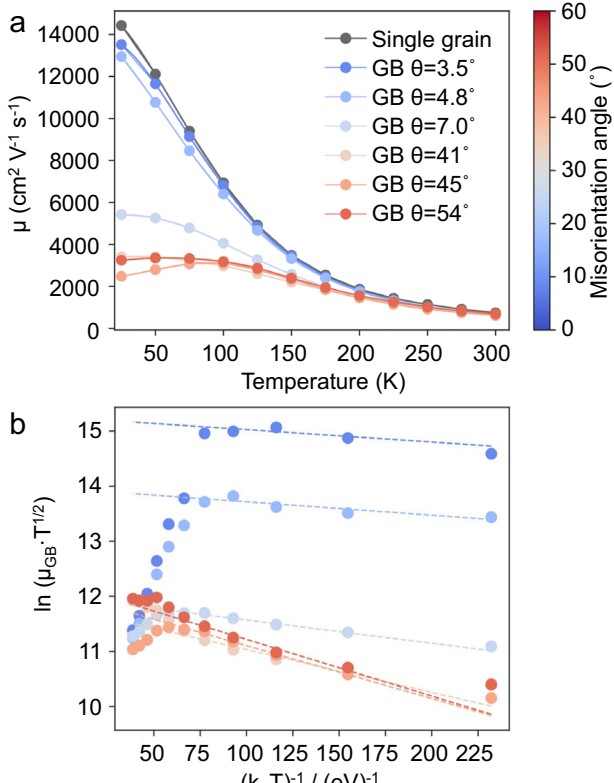

**Fig. 2 | Temperature-dependent electrical transport properties measured on the lamellar samples. a** Carrier mobility $\mu$ for samples with individual GBs that differ in their misorientation angle ($\theta$). A sample with a single grain is measured for reference. **b** The GB potential barrier height can be obtained upon fitting the low-temperature data for the temperature dependence of the mobility plotted as $ln(\mu_{GB}T^{1/2})$ versus $(1/k_BT)$, where $\mu_{GB}$ stands for the mobility of charge carriers at the GB area obtained according to Eq. (1).

grain interior or at GBs. However, the absence of such a scattering phenomenon in the single-grain sample implies that the scattering strength of carriers by charged defects in the grain interior is much weaker than in the GB area.

Upon the reasonable assumption that the scattering of GBs and phonons is uncorrelated, the measured carrier mobility $\mu$ will obey the following relationship[28]

$$\mu^{-1} = \mu_G^{-1} + \mu_{GB}^{-1} \qquad (1)$$

where $\mu_G$ is the mobility of the single grain, while $\mu_{GB}$ is the mobility of the carriers at the GB area. Note that both the acoustic and polar optical phonon scattering leads to a decrease in mobility with increasing temperature[29], which is incompatible with the observed thermally activated behavior in the specimens including GBs. For GB-dominated carrier scattering, the mobility can be expressed as[28]

$$\mu_{GB} = Le\left(\frac{1}{2\pi m^* k_B T}\right)^{\frac{1}{2}} \exp\left(\frac{-E_b}{k_B T}\right) \qquad (2)$$

Here, $L$ is the grain size, $e$ is the electron charge, $m^*$ is the effective mass of charge carriers, $k_B$ is the Boltzmann constant, $T$ is the temperature, and $E_b$ is the potential barrier height. In our specimen geometry for PPMS measurements (Fig. 1d), the grain size $L$ for all lamellar samples is the same and equals half of the distance between two voltage electrodes (2 μm in our case). Therefore, the GB carrier mobility is mainly influenced by the potential barrier

height. Equation (2) can thus be rewritten as

$$\ln\left(\mu_{GB}T^{\frac{1}{2}}\right) = \frac{-E_b}{k_B T} + \text{Constant} \qquad (3)$$

Figure 2b plots the profiles of $\ln(\mu_{GB}T^{\frac{1}{2}})$ versus $\frac{1}{k_B T}$, from which the potential barrier height ($E_b$) of GBs can be determined as the negative slope. The barrier height of HAGBs is about three to five times larger than that of LAGBs. This explains the distinctively different temperature dependence of the carrier mobility at low temperatures, where scattering at GBs dominates. While the carrier mobility is less affected near room temperature, the amplitude of the total scattering strength at the GB scales with the average grain size (Eq. 2). The impact of charge carrier scattering at GBs increases significantly for nanostructured compounds even at elevated temperatures. For example, in a polycrystalline iodine-doped PbTe at 300 K, the electrical conductivity decreased from about 6000 S/cm for a grain size of 3 μm to about 1000 S/cm for a grain size of 200 nm at 300 K[30].

The impact of the GB can be quantified within the trapping state model[31], where a large number of defects and the free volume at the boundary result in the formation of trapping states that capture free carriers. In this model, the GB is electrically charged by trapped carriers leading to a depletion region in its vicinity[32]. A potential barrier develops due to the charge redistribution, which imposes an obstacle to the motion of charge carriers from one grain to another. The trapping state model describes the GB potential barrier height $E_b$ as[31,32]

$$E_b = \frac{e^2 Q_t^2}{8N\varepsilon_{st}} \qquad (4)$$

where $Q_t$ is the density of trapping states at the GB, $N$ is the concentration of ionized impurity atoms (usually dopant atoms), and $\varepsilon_{st}$ is the static dielectric permittivity. The $N$ value can be regarded as the same for all samples studied in this work given their very similar Hall carrier concentrations (Fig. S1) and dopant composition in the matrix measured by APT (will be discussed below). Therefore, $Q_t$ and $\varepsilon_{st}$ are the two factors that determine $E_b$, which will be separately discussed below.

**Trapping state density and chemical bonding mechanisms**

To unravel the possible contribution of $Q_t$ and $\varepsilon_{st}$ additional information is needed. Interestingly, it comes from an unexpected method. Figure 3a shows the distribution of Pb, Te, and Ag atoms in an APT specimen containing a LAGB. This LAGB consists of parallel dislocation arrays, as highlighted by iso-composition surfaces with a composition of Ag greater than 5 at%[33]. The average Ag composition within this LAGB plane can reach about 5 at%, as revealed by the 1D composition profile across the GB (Fig. 3b). The inset from the top view of the GB further depicts the segregation behavior of Ag at dislocation cores, while the regions between dislocation lines are not enriched in Ag. In contrast, Ag atoms cover the whole surface of the HAGB, which can be directly observed from the side and top views of the GB plane in Fig. 3c, d, respectively. The observation provides indirect evidence of more free volume and defects at the HAGB. The 1D composition profile across the HAGB in Fig. 3d also confirms this conclusion. It shows a stronger segregation degree with an average Ag composition of 10 at%. Note that the composition of Ag at GBs is far less than what is expected for thermodynamically stable $Ag_2Te$ precipitates or Ag metal. We can thus exclude the contribution of a second phase to charge carrier scattering. The Gibbsian interfacial excess of impurity atoms, determined by the method described by ref. [34], at the LAGB and HAGB, is 10.5 and 14.0 at/nm², respectively (see Fig. S2).

Besides the GB composition and trapping states, APT can also capture information concerning the chemical bonding mechanism at the GB and grain interior. APT works on the principle of

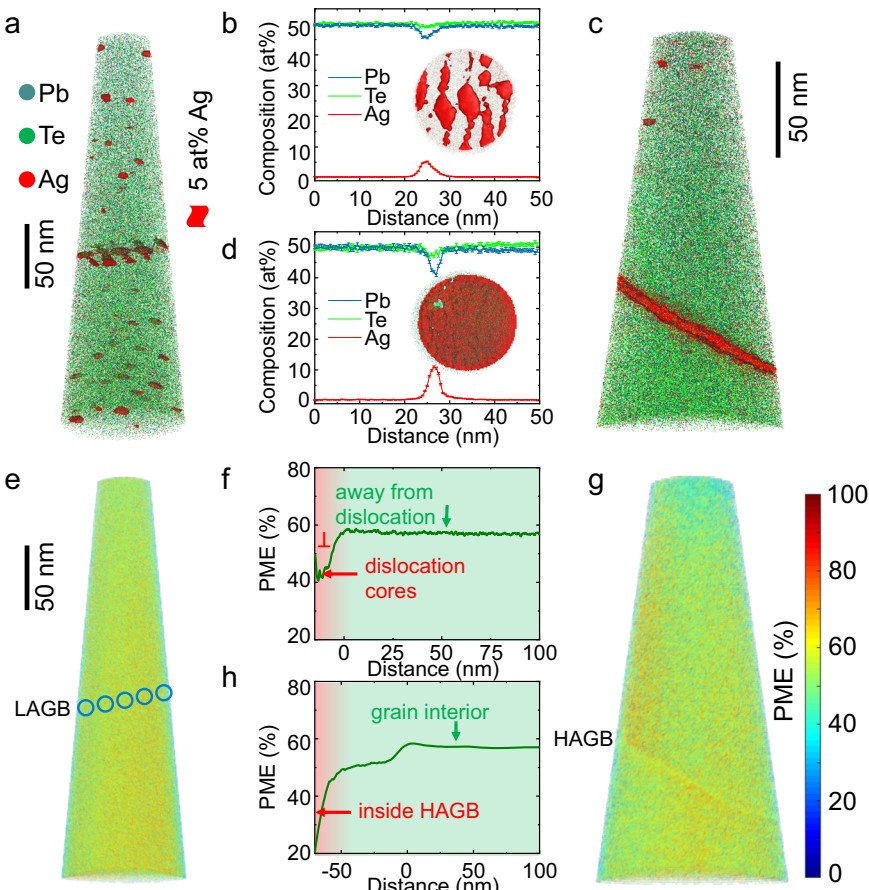

**Fig. 3 | Atom probe tomography investigation of two specimens including individual LAGB and HAGB taken from the Ag-doped PbTe. a** 3D map showing the distribution of Pb, Te, and Ag atoms. The iso-composition surfaces of 5 at% Ag depict the Ag-decorated dislocation arrays at a LAGB located in the middle of the specimen. **b** 1D composition profile across this LAGB showing Ag enrichment at the boundary. Inset: top view of this LAGB. **c** 3D reconstruction containing HAGB with Ag-rich feature at the boundary. **d** 1D composition profile across the HAGB shows a more pronounced Ag enrichment as compared to LAGB. Inset: top view of this HAGB. Ag impurities cover the whole GB plane, indicative of more available trapping states. **e** PME map of the LAGB sample, where the edges of dislocations are indicated by circles by referring to the positions on the composition map. **f** PME proximity histogram from inside the dislocation cores (below 0 nm) to the surrounding matrix (above 0 nm) calculated by defining iso-composition surfaces of Ag ions that depict the Ag-rich dislocations. This proximity histogram proves the low PME value inside the dislocation cores. **g** PME map of the HAGB sample, where a slight contrast change can be observed at the HAGB. **h** PME proximity histogram from inside the HAGB (below 0 nm) to the outside grain interior (above 0 nm) shows a significant drop of PME inside the HAGB while the grain interior shows a relatively high PME value, corroborating its MVB nature.

high-field-induced ionization and evaporation of atoms upon breaking chemical bonds[35]. It has been proven in many studies that an abnormal bond rupture with an unusually high probability to form several distinct fragments upon a single successful laser pulse, also called "probability of multiple events (PME)", in APT measurements is characteristic of materials utilizing metavalent bonding (MVB)[36,37]. More detailed discussions can be found in Fig. S3. MVB is a fundamental chemical bonding mechanism in solids that distinctively differs from covalent, ionic, and metallic bonding[38–41]. Since the bond rupture observed by APT is characteristic of MVB, this technique enables a local determination of the prevalent bonding mechanism. Figure 3e shows the 3D PME map of the LAGB sample; now possible thanks to the home-coded software EPOSA[42]. A high PME value (~60%) of the matrix confirms the MVB nature of PbTe, in agreement with previous studies[43]. More interestingly, the PME proximity histogram calculated by defining ionic iso-concentration surfaces reveals the PME changes from the dislocation cores to the surrounding matrix even though the PME map does not show significant differences due to the contrast superposition with the surrounding matrix in 3D space. Figure 3f shows a high PME value of 60% for the regions around dislocations while a lower PME of 40% is found inside the dislocation cores. Similarly, the PME proximity histogram of the HAGB indicates a strong

reduction of PME down to 20% within the HAGB plane, Fig. 3h. The much lower PME value is solid evidence of the breakdown of MVB within the dislocation cores and the HAGB. Note that the bond-rupture process in APT is sensitive to the type of chemical bonding rather than the arrangement of neighboring atoms. This can be seen when comparing the bond rupture between amorphous and crystalline phase change materials. While the amorphous phase is characterized by covalent bonding and hence reveals a low PME, the crystalline state employs MVB and thus shows a high PME[36]. This emphasizes that both the microstructure and chemical bonding mechanisms should be considered when describing the GBs of metavalent solids. In particular, the resulting change of material properties in the vicinity of the GB needs to be considered, as will be discussed in detail below.

The degree of segregation of Ag to GBs is a consequence of the increased density of trapping states. These trapping states attract both charge carriers and dopants. According to the Langmuir-McLean theory[44], for the minimum value of the GB free energy ($\triangle G$), the adsorption of impurities at GB can be described as

$$\frac{X_b}{X_b^0 - X_b} = \frac{X_c}{1 - X_c} \exp\left(\frac{-\triangle G}{RT}\right) \tag{5}$$

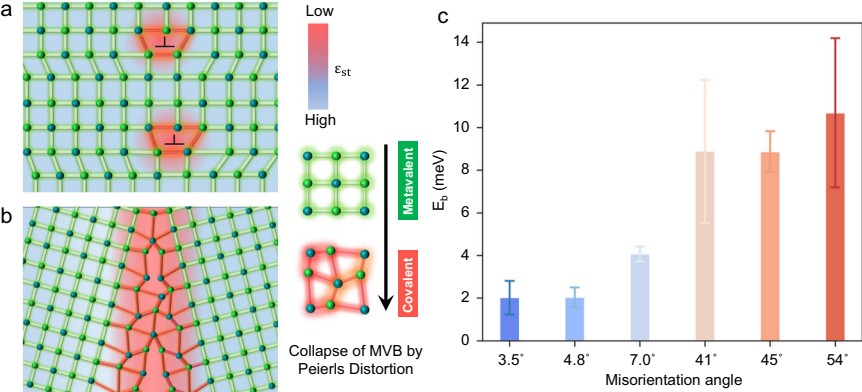

**Fig. 4 | Schematics illustrating a possible atomic arrangement causing the collapse of MVB at GBs. a** A sketch of a LAGB containing two dislocations. The MVB nature is destroyed at the dislocation core due to a Peierls distortion and the concomitant reduced effective coordination number, accompanied by a lower value of $\varepsilon_{st}$. **b** A sketch of a HAGB, where a much stronger disorder can be observed because dislocation arrays cannot accommodate the large mismatch anymore. The distortions within the HAGB destroy MVB in a larger region. The background color of these two figures indicates the value of the static dielectric constant, which is correlated with the chemical bonding mechanism. MVB is characterized by high $\varepsilon_{st}$ values, while its destruction leads to low $\varepsilon_{st}$ values. **c** Histogram of the GB potential barrier showing the increased charge carrier scattering at GBs from LAGB to HAGB due to the weakened dielectric screening effect as explained on the left side.

This equation demonstrates that for a GB with more trapping states available for segregated atoms at saturation ($X_b^0$), the fraction of actual sites covered with segregated atoms ($X_b$) is also larger, given that the bulk solute molar fraction ($X_c$) is constant. This formula relates the segregation density and the density of trapping states. Thus, we can compare the density of trapping states in different GB structures by determining the GB Gibbsian excess. Figure 3b, d demonstrate more pronounced segregation and thus a higher number of available trapping states over the HAGB compared to the LAGB. The Gibbsian excess of impurity Ag atoms (Fig. S2) corresponds to roughly 1.3–1.5 times higher number of trapping states for the HAGB than that for the LAGB. This will lead to a difference in the GB barrier height by a factor of about 2 according to Eq. (4). Yet, this factor alone cannot explain the observed increase of the $E_b$ value by a factor of about 4–5 from LAGBs to HAGBs.

Equation (4) indicates that the static dielectric constant $\varepsilon_{st}$ also impacts the $E_b$ value. The parameter $\varepsilon_{st}$ is defined as the ratio between the constant electric field within a material and the corresponding electric displacement in the DC limit. $\varepsilon_{st}$ is the low-frequency limit of the dielectric function, i.e., the response of a sample to an applied electric field. This dielectric function has a few features that are characteristic of metavalent solids. These characteristics are closely related to the weak bonding in metavalent solids, leading to soft phonons and pronounced polarizability of electrons. In particular, the valence electrons in MVB compounds such as PbTe are rather delocalized due to the large overlap between p-electrons. This causes a strong polarizability and a concomitant large $\varepsilon_\infty$. The large value of $\varepsilon_\infty$ is one of the fingerprints of MVB[38,45,46]. Yet, MVB compounds also show a unique property portfolio including a high Born effective charge, which leads to a large value of the static dielectric constant ($\varepsilon_{st}$). According to the Lyddane-Sachs-Teller-Relation[47], the ratio of both quantities

$$\frac{\varepsilon_{st}}{\varepsilon_\infty} = \frac{\omega_{LO}^2}{\omega_{TO}^2} \qquad (6)$$

is related to $\omega_{LO}$ and $\omega_{TO}$, the frequency of longitudinal and transverse optical phonons, respectively. It is well-known that MVB compounds show a very low frequency of transverse optical modes, evidence of near-by instability. This explains why these materials show extremely high values of $\varepsilon_{st}$. For example, the static dielectric constant of PbTe is 380, which is about 30 times that of Si ($\varepsilon_{st} = 13$)[32].

The static dielectric constant ($\varepsilon_{st}$) is important to reduce the GB scattering as it leads to dielectric screening[48]. The very large $\varepsilon_{st}$ of PbTe gives rise to a strong screening of charged defects. This explains why no ionized impurity scattering is observed in the single-grain sample (Fig. 2a) although Ag dopants are found in the matrix. Otherwise, the carrier mobility of the single-grain sample should show a temperature dependence of $T^{1.5}$ due to the increased mean electron velocity with the increasing temperature that weakens the charge in momentum through the Coulomb interaction between charge carriers and ionized impurities[49]. This also explains the high charge carrier mobility of PbTe. In contrast, the carrier mobility in P- or B-doped Si decreases significantly with increasing the doping level due to the strongly ionized impurity scattering[50]. Yet, a potential barrier is observed for PbTe samples including GBs, which is indicative of a reduced dielectric screening ability within the GB area. This is consistent with the APT findings that the PME value drops (MVB collapse) inside the dislocation cores and HAGBs. Once MVB breaks down, the static dielectric constant will decrease significantly, causing increased impurity scattering.

## Mechanisms of the chemical bonding transition at the grain boundary

Figure 4a provides a possible microscopic picture of the atomic arrangement within the GB region based on the chemical bonding information obtained by APT, in line with the scenario discussed above. For a perfect PbTe crystal, each atom has octahedral coordination. Yet, the number of valence electrons in p-orbitals forming σ-bonds per atom is just three. Hence, for adjacent atoms only half of an electron pair, i.e., in total one electron is available to form a bond. This is different from the picture of a covalent bond developed by Lewis which involves the sharing of two electrons to form one electron pair between adjacent atoms[51]. This half-filled σ-bond is the hallmark of MVB. However, this atomic arrangement likely changes at the dislocation core, presumably leading to pronounced distortions away from perfect octahedral coordination. This distortion is frequently denoted as a Peierls distortion[39,52], which causes a redistribution of electrons between short and long bonds. The short bonds will become stronger with a larger number of shared electrons while the longer bonds evolve the opposite[40]. A strong degree of Peierls distortion will drive the transition from MVB to covalent bonding, as has been demonstrated in the material systems of GeTe-GeSe and $Sb_2Te_3$-$Sb_2Se_3$[39]. Since the atomic bonds are only modified at the dislocation core for LAGBs, the area between the dislocation cores remains almost the same configuration as the matrix and thus the MVB mechanism survives. Different from the LAGB, the dislocation spacing gets smaller with increasing

misorientation and can no longer be accommodated at the HAGB. The whole HAGB plane is distorted with a large free volume. Thereby, presumably, a pronounced Peierls distortion exists in the vicinity of the HAGB, leading to the collapse of MVB (Fig. 4b), consistent with the observed decrease of PME at the HAGB. As has been explained above, the large degree of electron delocalization and polarization leads to the high values of $\varepsilon_\infty$ and $\varepsilon_{st}$. The collapse of MVB will lead to a reduction of $\varepsilon_{st}$ by a factor of about 2–3 as can be seen from the sharp decrease in $\varepsilon_\infty$ and Born effective charge upon the transition from MVB to covalent bonding[39]. This turns out to cause an increase in GB barrier height by a factor of about 2–3 from LAGBs to HAGBs. In conjunction with the trapping states scattering at GBs, a much larger GB potential barrier height is obtained in HAGBs as illustrated in Fig. 4c.

## Discussion

In summary, a novel correlative method has been developed to investigate the relationship between microstructure, transport properties, composition, and bonding mechanism for an individual GB using EBSD-PPMS-APT procedures. With this approach, we have demonstrated that charge carriers can be scattered more strongly at the HAGBs than at the LAGBs in an Ag-doped PbTe compound. The GB potential barrier height is about 8–10 meV for HAGBs, while it is only 2 meV for LAGBs. APT measurements reveal a 1.3 times larger degree of Gibbsian excess of Ag atoms at the HAGB than that at the LAGB, indicating a larger fraction of trapping states at the HAGB. Moreover, APT also probes the drop of PME value at the whole plane of HAGB but only within the dislocation cores of LAGB. This indicates that MVB is completely destroyed at the HAGB while only locally destroyed around dislocation cores at the LAGB. It is well-known that MVB is characterized by a high value of both static and optical dielectric constants. The collapse of MVB will reduce the dielectric constant and thus the dielectric screening ability. As a consequence, both the larger number of trapping states and the complete breakdown of MVB at HAGBs cause strong charge carrier scattering. In contrast, the charge carriers will only be slightly scattered at dislocation cores of LAGBs and thus show much higher mobility. This finding provides a microscopic explanation of why thermoelectric properties can be improved by introducing dislocations, which lower the lattice thermal conductivity yet maintain the electrical conductivity, as have already been implemented in several MVB thermoelectrics[6,18,53,54]. As there is a significant number of materials utilizing MVB, our findings on charge transport across GBs over a broad temperature range are important for various applications including power generation (as thermoelectric), memory (as phase change material), mid-infrared lasers, detectors, and photovoltaics[55].

## Methods

### Synthesis

The raw materials of Pb powders (99.96%, Riedel-de Haen), Te ingots (99.99%, Strem Chemicals), and Ag ingots (99.999%, Alfa Aesar) were weighed according to the nominal composition of PbTe- 0.5% $Ag_2Te$ and sealed in quartz tubes under dynamic vacuum. The sealed tube was heated to 1273 K in 12 h and dwelled for another 6 h in a vertical programmable tube furnace to ensure a homogeneous melting of the mixture. After that, the melt was rapidly cooled down by quenching in ice water. The obtained ingot was then hand-ground to fine powders with an agate mortar and pestle in an Ar-filled glove box. The resulting powders were consolidated by spark plasma sintering at 923 K for 30 min under an axial pressure of 45 MPa. In this work, we choose Ag as the dopant because it has been widely used in thermoelectrics and it can often decorate GBs, which is important for the visualization of different GB features in the APT results. The content of Ag should be low (PbTe-0.005 Ag) to avoid the formation of complex structural defects and second-phase precipitates at GBs[25].

### Fabrication of the measurement device with nano-electrodes

The prepared nano-electrodes with the hall-bar structure on $SiO_2$/Si substrate are used for connecting lamellae with a single GB or grain to measure the electrical transport properties. Electron-beam lithography (EBL) and e-beam evaporation (Ti/Au 5/25 nm) were used to define these electrode contacts. The distance between two voltage electrodes is 4 μm. The dimensions for the transferred lamella are 2–3 μm in width, 1 μm in thickness, and 12 μm in length. The sizes of the contacts with the sample for voltage and current electrodes are 1 μm and 200 nm, respectively. The present Au electrodes are very stable for the measurement below room temperature. Yet, our design should also be adapted to a high-temperature measurement device. In this case, also the devices need to be modified. The narrow and thin Au electrodes are prone to inter-diffusion and chemical reactions at elevated temperatures, which could cause an open circuit. A possible solution is to replace Au with Pt and make the electrodes thicker because Pt is more stable than Au.

### Structural characterizations

The sample was first mechanically polished with SiC abrasive papers by increasing the mesh number from 800 to 4000 step by step. And then $SiO_2$ suspension liquid with a particle size of 50 nm was used to do the fine polishing to get a high-quality surface. The electron backscatter diffraction (EBSD) and transmission Kikuchi diffraction (TKD) measurements were conducted at an acceleration voltage of 20 kV and a beam current of 1.6 nA with an inclination angle of 70° and 52°, respectively, using the Helios NanoLab 650 equipped with Hikari high-speed camera (EDAX/TSL). The EBSD and TKD data were processed by the TSL OIM Analysis 7.0 software.

### Site-specific "lift-out" to prepare micro-scale lamellar samples

As briefly described in the main text, this process aims at lifting out a specific GB characterized by EBSD. The whole process was finished using a dual-beam focused ion beam (FIB, Helios NanoLab 650, FEI). The GB was first aligned vertically in both SEM and FIB windows and then the stage was tilted to 52°, perpendicular to the FIB column. Three rectangular areas were first cut ("U" cut) by FIB at a voltage of 30 kV and a beam current of 0.79 nA. And then the stage was tilted back to 0° and the bottom part was cut through. The micromanipulator (Omniprobe) was inserted to attach the lamella and they were glued by the Pt gas injection system. After that, the lamella was cut free and then transferred to a Cu grid for shaping the width and thickness with high-quality surfaces by low voltage FIB cross-section cleaning. The Ga contamination for FIB-prepared specimens depends on the acceleration voltage and current. For example, it has been demonstrated that the penetration depth of Ga in Si is about 40 nm at 30 kV, while it decreases to about 5 nm and <1 nm by cleaning the final surface with an acceleration voltage of 5 and 2 kV, respectively[26]. We used a very low voltage (2 kV) and current (4 pA) to clean the surface of specimens at the final step. Thus, the Ga contamination should be marginal. Moreover, we did not observe Ga signals in the APT data, which further demonstrates that the content of Ga in the PbTe specimens for both PPMS and APT measurements can be neglected. The well-shaped lamella was finally mounted onto the measurement device fabricated by EBL and glued by Pt.

### APT sample preparation, measurement, and data analysis

Needle-shaped APT specimens were prepared by the standard "lift-out" method[26] but mounted on a half-grid of Mo for TKD characterization. Low voltage cleaning (2 kV) was used at the final step to remove surface contamination and amorphization. The APT measurements were performed on a local electrode atom probe (LEAP 4000X Si, Cameca) in a laser mode with UV light (wavelength = 355 nm). The corresponding laser pulse energy is 20 pJ with a pulse repetition rate of 200 kHz. The average detection rate is

1%; the base temperature of the specimen is 40 K; the flight path of ions is 160 mm. The detection efficiency is 50% limited by the open area left between the microchannels on the detector plates. The APT data were processed using the commercial software package IVAS 3.8.0.

### Electrical transport property measurements

To confirm ohmic contacting in these devices, the I-V curves were measured with a DC source meter (Keithley 2400) and digital multimeter. The electrical transport measurement was performed with a Dynacool from Quantum Design using a driving current of 1 μA to avoid the Peltier effect and Joule heating. The resistivity was determined by measuring the longitudinal resistance of samples cooled from 300 K to 10 K. For hall measurements, a magnetic field ranging from −1 to 1 T was utilized.

### Reporting summary

Further information on research design is available in the Nature Portfolio Reporting Summary linked to this article.

## Data availability

The authors declare that the data supporting the findings of this study are available on reasonable request.

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

## Acknowledgements

Y.Y., O.C.-M., and M.W. acknowledge the financial support from DFG SFB 917 project. Y.Y. acknowledges the financial support under the Excellence Strategy of the Federal Government and the Länder within the ERS RWTH Start-Up grant (Grant No. StUpPD_392-21). C.Z. acknowledges the financial support from Fundamental Research Funds for the Central University (D5000220051). M.Wegener. from RWTH Aachen University is acknowledged for his help in preparing the lamella samples for PPMS measurements.

## Author contributions

Y.Y. conceived the project. O.C-M. and M.W. supervised the project. R.W. performed the EBL and PPMS experiments, analyzed the transport data under the supervision of Y.Y., and draw the sketches. Y.Y. and S.J. performed the EBSD scanning, prepared the lamellae and APT specimens, and carried out the APT characterization and data analyses. C.Z. provided the Ag-doped PbTe sample and helpful discussions. Y.Y. wrote the draft. O.C-M. and M.W. edited the draft. All authors commented on and approved the submission of this work.

## Funding

## Competing interests

The authors declare no competing interests.
