## [Peer Review File · Nature Communications]

Strong Charge Carrier Scattering at Grain Boundaries of PbTe caused by the Collapse of Metavalent BondingREVIEWER COMMENTS

Reviewer #1 (Remarks to the Author):

The present manuscript describes a very ingenious method to correlate the local electrical transport properties across a specific grain boundary with its microstructures and chemical bonding. They found that the grain boundary potential barrier height for charge transport is larger at high-angle grain boundaries than that at low-angle grain boundaries. A trapping state model is used to understand the origin of the potential barrier. Two parameters, i.e., the trapping state density and the static dielectric constant, are revealed to be responsible for the difference in barrier height for different types of grain boundaries. Experimentally, it is very difficult to study these two parameters. Interestingly, the authors discussed the contribution of each parameter to the barrier height according to the results of atom probe tomography. They conclude that the enlarged barrier height at high-angle grain boundaries is due to the joint effect of a higher density of trapping states and the collapse of metavalent bonding. The impact of trapping states is quite well known but I found the argument on the chemical bonding very interesting and also convincing according to the well-designed experiments.

Thus, I would like to recommend the publication of this work in Nature Communications with a minor revision.

It would be good if the authors can clarify the following comments in the revision because they are not very clear to me at the present stage.

1. The authors illustrate the change of chemical bonding mechanisms in Figure 4 according to the APT results in Figure 3. The logical correlations between these two figures seem not very clear. Please clarify and improve it.
2. The authors focus on grain boundary scattering in this work. Will the ionized impurity scattering also play a role? If yes, how to exclude the impact of ionized impurity scattering? Similarly, how about the other mechanism, such as dislocation, acoustic or optical phonon scattering? Concerning this, the literatures on PbTe in the year 2022 should be cited.
3. The bar-shape samples were prepared by Ga-ion FIB. It is well known that Ga can penetrate the materials and thus behave as dopants. Will the Ga contamination influence the results and conclusion since Ga is a good dopant in PbTe?
4. The authors studied the charge transport across the grain boundary in PbTe below room temperature. However, PbTe is typically used at high temperatures for thermoelectrics. Can the authors measure the high-temperature properties? If not, what are the difficulties, and do they have a possible solution?
5. There is a typo in Figure 4. The authors use “ ϵ_{st} ” in the text to describe the static dielectric constant but use “ ϵ_s ” in the Figure. Also other typos should be checked carefully.

Reviewer #2 (Remarks to the Author):

The study of grain boundaries (GBs) is of great significance in materials science. GBs impact many aspects of the properties of materials such as the mechanical and functional properties. Although the microstructures, compositions, and even the transport properties of GBs have been extensively studied, the correlation of multi-dimensional features for one individual GBs remains a challenge. This work creatively developed a method to investigate the origin of the GB potential barrier for charge transport at an individual GB. More interestingly, they found that the GB barrier height is linked to the change in local chemical bonding. Atom probe tomography was employed to characterize the defects and chemical

bonding mechanisms at the GB. They systematically studied several different types of GBs in Ag-doped PbTe, which shows outstanding thermoelectric properties. The results and conclusions achieved in this work enrich the understanding of charge carrier scattering at GBs. Thus, I recommend to accept this manuscript. However, some concerns should be addressed as listed below.

1. The authors studied the low-temperature transport properties of PbTe. Yet, this compound is often utilized as thermoelectrics at about 700 K. It seems that the charge carrier scattering at GBs is very weak and independent of the types of GBs above room temperature. What could be the significance of this work to the application of PbTe?
2. The authors described the change of chemical bonding mechanisms at the GB according to the atom probe tomography results. They claimed that the so-called PME value can be used to identify the collapse of metavalent bonding. What is the intrinsic causality between chemical bonding and PME?
3. Will the charge scattering at GBs in non-MVB materials differ from the MVB compounds? The collapse of metavalent bonding should only be valid in MVB compounds.

Reviewer #3 (Remarks to the Author):

Recently, the effect of grain boundaries has attracted increasing attentions in thermoelectric community. A direct observation of how the grain boundary affect transport properties are challenging. In this work, a direct measurement on how a single grain boundary affect electrical transport property is realized. This is the highlight of this manuscript. The results are in detail and the discussion are reasonable. Therefore, I think this manuscript can be accepted for publication after addressing following questions.

1. How to ensure the accuracy of determining GB potential barrier height? In Figure 2B, the three red data apparently show larger slope than three blue data. But for three red data, I think it is hard to tell who has larger negative slope, especially for $\theta = 41^\circ$ and $\theta = 54^\circ$. It seems that the result largely depends on how the author fit the data. If so, the reported exact value of E_b might be misleading for readers. Any comment or supplement on this issue?
2. Will the grain boundary scattering be related to carrier concentration (Fermi level)? I notice that the samples in this work have carrier concentration much lower than the optimization for thermoelectric performance. Is this an intentional choose? Some explanation or discussion are suggested.
3. The authors said MVB compounds show high dielectric constants. Is this a rigorous conclusion? Or, PbTe is just one of special cases? I think more discussion on this issue is necessary. This is critical for the main conclusion of the manuscript.
4. Following question 3. I can understand the logic that a reduced dielectric screening ability leading to lower carrier mobility in GB due to MVB collapse. However, will this logic be contradicted to the case that lots of covalent compounds (such as Si and InSb) usually show high carrier mobility than MVB compounds?
5. The scope of current title is too large. I suggest the material PbTe to be clarified in the title.

REVIEWER COMMENTS

Reviewer #1 (Remarks to the Author):

The present manuscript describes a very ingenious method to correlate the local electrical transport properties across a specific grain boundary with its microstructures and chemical bonding. They found that the grain boundary potential barrier height for charge transport is larger at high-angle grain boundaries than that at low-angle grain boundaries. A trapping state model is used to understand the origin of the potential barrier. Two parameters, i.e., the trapping state density and the static dielectric constant, are revealed to be responsible for the difference in barrier height for different types of grain boundaries. Experimentally, it is very difficult to study these two parameters. Interestingly, the authors discussed the contribution of each parameter to the barrier height according to the results of atom probe tomography. They conclude that the enlarged barrier height at high-angle grain boundaries is due to the joint effect of a higher density of trapping states and the collapse of metavalent bonding. The impact of trapping states is quite well known but I found the argument on the chemical bonding very interesting and also convincing according to the well-designed experiments. Thus, I would like to recommend the publication of this work in Nature Communications with a minor revision. It would be good if the authors can clarify the following comments in the revision because they are not very clear to me at the present stage.

General Response: We sincerely appreciate the positive evaluation of our work. It is pleasing to read that our work is considered novel and significant. We also thank the reviewer for the constructive comments and suggestions. Each comment has been addressed point-by-point below and the corresponding revisions were made in the revised manuscript as highlighted in yellow.

1. The authors illustrate the change of chemical bonding mechanisms in Figure 4 according to the APT results in Figure 3. The logical correlations between these two figures seem not very clear. Please clarify and improve it.

Response: Figure 3 shows information about the microstructure and chemical bonding mechanism at the low-angle grain boundary (LAGB) and the high-angle grain boundary (HAGB). The linear features depicted by the Ag-rich iso-composition surfaces in Figure 3A are dislocations, as have been observed in other APT data [Yu et al., *ACS Appl. Mater. Interfaces* 2018, 10, 3609-3615; Kuzmina et al., *Science* 2015, 349, 1080]. This demonstrates that the LAGB can be described by dislocation arrays. In contrast, the HAGB shows a smooth spread of Ag covering the whole GB plane. This is because, for a random HAGB, there are more free volumes and trapping states for the segregation of impurity atoms. This structural information provides the basis for the sketches of atomic models in Figure 4.

Besides the microstructures, Figure 3 reflects changes in chemical bonding at the GBs. This correlation is derived based on the bond-breaking behavior in APT. APT works on the principle of high-field-induced ionization and evaporation of atoms by breaking chemical bonds [Gault et al., *Nat. Rev. Methods Primers* 2021, 1, 51]. In general, the evaporation process triggered by a successful laser pulse primarily emits a single ion, which is called a single event. This is valid for many materials including metals, as well as covalent and ionic compounds. Interestingly, Zhu et al. found that a successful laser pulse often dislodges more than one fragment in a collection of materials that utilize metavalent bonding [Zhu et al., *Adv. Mater.* 2018, 30, 1706735]. Thus, a large value of “probability of multiple events (PME)” is a hallmark of metavalently bonded solids and can be utilized to identify the existence or collapse of metavalent bonding. Figure 3 shows the collapse of metavalent bonding at dislocation cores and HAGB by the drop in the PME value. Moreover, the bond-rupture process is sensitive to the type and strength of chemical bonding rather than the arrangement of neighboring atoms, as one can see from striking differences in the bond rupture between amorphous and crystalline phase change materials. Thus, different chemical bonding mechanisms at the GBs are also sketched in Figure 4.

Revisions: The changes are highlighted in yellow in the revised manuscript on pages 7, 8, and 10.

2. The authors focus on grain boundary scattering in this work. Will the ionized impurity scattering also play a role? If yes, how to exclude the impact of ionized impurity scattering? Similarly, how about the other mechanism, such as dislocation, acoustic or optical phonon scattering? Concerning this, the literature on PbTe in the year 2022 should be cited.

Response: Both the ionized impurity scattering and the GB scattering share similar origins that stem from the Coulomb force between electrons and charged defects. In general, the ionized impurity scattering describes the homogeneous scattering event within the (doped) grain. In contrast, the scattering event localized at the GB caused by the collection of charged defects forming a space charge layer is attributed to GB scattering. If the ionized impurity scattering dominates, the carrier mobility of the single-grain sample should also be impacted and would show a temperature dependence of $T^{1.5}$, due to the increased mean electron velocity with the increasing temperature that weakens the change in momentum through the Coulomb interaction [Conwell et al., *Physical Review* 1950, 77, 388-390]. Yet, this is not the case as observed in Figure 2A. Thus, we can exclude the impact of ionized impurity scattering even though dopants are found in the grain interior. This is also a consequence of the strong dielectric screening of PbTe within the grain due to its metavalent bonding. This conclusion can be rationalized due to the high value of the static dielectric constant ϵ_{st} , a feature that is closely linked to MVB in PbTe. We did not observe any dislocations in the grain interior but only at the LAGB by APT. Our results show that charge carriers are only weakly scattered by dislocations according to the rather small potential barrier height at LAGBs. This is an important indication that the thermoelectric properties can be improved by introducing dislocations, which lower the lattice thermal conductivity yet maintain the electrical conductivity, as have already been implemented in several MVB thermoelectrics

[Kim et al., *Science* 2015, 348, 109-114; Chen et al., *Adv. Mater.* 2017, 29, 1606768; You et al., *Energy Environ. Sci.* 2018, 11, 1848-1858; Xu et al., *Adv. Mater.* 2022, 34, 2202949].

The acoustic phonon scattering dominates the charge transport at high temperatures, which is responsible for the reduced carrier mobility with increasing temperature. The polar optical phonon scattering also leads to a decrease in mobility with temperature [Wang et al., *Adv. Energy Mater.* 2013, 3, 488-195]. Thus, it cannot explain the thermally activated behavior at low temperatures in the GB-containing specimens. Moreover, these two scattering mechanisms impact the charge transport in the grain interior. The GB potential barriers add an extra scattering source for the samples including GBs. It should be emphasized that the impact of GB scattering on carrier mobility is obtained by subtracting the contribution of all other scattering mechanisms included in the single-grain sample. In this regard, our analyses on the GB potential barrier height have excluded other potential interferences.

Revisions: We added more discussions on the ionized impurity scattering on page 10. We also extended our outlook on the design of materials based on the weak scattering effect of dislocations on charge transport, see page 12. We clarified that the scattering mechanisms such as acoustic phonon and polar optical phonon scattering to charge carriers are excluded by subtracting the data for the single-grain sample from the measured Hall mobility for GB-included samples, see page 5. We also updated the references.

3. The bar-shape samples were prepared by Ga-ion FIB. It is well known that Ga can penetrate the materials and thus behave as dopants. Will the Ga contamination influence the results and conclusion since Ga is a good dopant in PbTe?

Response: The Ga contamination for FIB-prepared specimens depends on the acceleration voltage and current. It has been demonstrated that the penetration depth of Ga in Si is about 40 nm at 30 kV, while it decreases to about 5 nm and <1 nm by cleaning the final surface with an acceleration voltage of 5 and 2 kV, respectively [Thompson et al., *Ultramicroscopy* 2007, 107, 131-139]. We used a very low voltage (2 kV) and current (4 pA) to clean the surface of specimens at the final step. Thus, the Ga contamination should be marginal. Moreover, we did not observe Ga signals in the APT data, which further demonstrates that the content of Ga in the PbTe specimens for both PPMS and APT measurements can be neglected.

Revisions: We added more details and discussions in the “Methods” part of the revised manuscript on page 14 to clarify the negligible influence of Ga contamination.

4. The authors studied the charge transport across the grain boundary in PbTe below room temperature. However, PbTe is typically used at high temperatures for thermoelectrics. Can the authors measure the high-temperature properties? If not, what are the difficulties, and do they have a possible solution?

Response: At present, we cannot measure the high-temperature properties since our PPMS system is only able to measure at room temperature and below. Yet, it is possible to perform such a measurement at high temperatures by employing a high-temperature setup. In this case, also the devices need to be modified. The narrow and thin Au electrodes are prone to interdiffusion and chemical reactions at elevated temperatures, which could cause an open circuit. A possible solution is to replace Au with Pt and make the electrodes thicker because Pt is more stable than Au. The reviewer's insightful comments have accelerated our efforts to improve the measurement samples. Nevertheless, the data and conclusions presented here already demonstrate the potential of the approach presented.

Revisions: We provided an alternative solution for the design of nano-electrodes for high-temperature measurements in the Section "Fabrication of the measurement device with nano-electrodes", see page 13.

5. There is a typo in Figure 4. The authors use " ϵ_{st} " in the text to describe the static dielectric constant but use " ϵ_s " in the Figure. Also, other typos should be checked carefully.

Response: Thank you for your careful reading. We have corrected the typo in Figure 4 and checked for other typos carefully.

Reviewer #2 (Remarks to the Author):

The study of grain boundaries (GBs) is of great significance in materials science. GBs impact many aspects of the properties of materials such as the mechanical and functional properties. Although the microstructures, compositions, and even the transport properties of GBs have been extensively studied, the correlation of multi-dimensional features for one individual GBs remains a challenge. This work creatively developed a method to investigate the origin of the GB potential barrier for charge transport at an individual GB. More interestingly, they found that the GB barrier height is linked to the change in local chemical bonding. Atom probe tomography was employed to characterize the defects and chemical bonding mechanisms at the GB. They systematically studied several different types of GBs in Ag-doped PbTe, which shows outstanding thermoelectric properties. The results and conclusions achieved in this work enrich the understanding of charge carrier scattering at GBs. Thus, I recommend to accept this manuscript. However, some concerns should be addressed as listed below.

General Response: We thank the reviewer for the positive and insightful comments on this work. Below you will find detailed answers to each suggestion or question.

1. The authors studied the low-temperature transport properties of PbTe. Yet, this compound is often utilized as thermoelectrics at about 700 K. It seems that the charge carrier scattering at GBs is very weak and independent of the types of GBs above room temperature. What could be the significance of this work to the application of PbTe?

Response: We appreciate this thoughtful comment. Indeed, the charge carrier scattering at GBs is very weak at about room temperature in our measurement device. Note that the contribution of GB scattering to the reduction in carrier mobility scales with the relative fraction of GB area. The grain size in our device is fixed at 2 μm . However, nanostructuring has been widely used in thermoelectrics. This would significantly increase the total magnitude of GB scattering on the reduced carrier mobility even at high temperatures. For example, in a polycrystalline iodine-doped PbTe, the electrical conductivity decreases from about 6000 S/cm for a grain size of 3 μm to about 1000 S/cm for a grain size of 200 nm at 300 K [Kishimoto et al., *Jpn. J. Appl. Phys.* 2003, 42, 501–508]. Our work demonstrates that sub-grains with small misorientations to neighboring grains could minimize the effect on electron scattering. This has been demonstrated in another MVB compound, Bi₂Te₃-Sb₂Te₃ alloy, by Kim et al. in 2015 [Kim et al., *Science* 2015, 348, 109-114]. Note that PbTe is just chosen as an example to demonstrate the origin of GB potential barriers. There are analogs to PbTe, which also employ metavalent bonding but show thermoelectric relevant applications near or below room temperature, such as Bi₂Te₃ and Bi-Sb alloy. The conclusions achieved in this work should also be valid for these low-temperature thermoelectric materials. Moreover, this work focuses on charge carrier transport, which is not confined to the area of thermoelectrics. The research methods developed in this work can be easily adapted to other solids that show important applications related to the charge transport properties such as phase-change materials and photovoltaics.

Revisions: We added more discussions on the scattering magnitude as a function of grain size on page 6.

2. The authors described the change of chemical bonding mechanisms at the GB according to the atom probe tomography results. They claimed that the so-called PME value can be used to identify the collapse of metavalent bonding. What is the intrinsic causality between chemical bonding and PME?

Response: This is another interesting question. We are currently collecting further data and working on a more extended explanation of this relationship. The understanding gained can be summarized as follows. The APT technique works on the principle of field evaporation theory. The surface atoms are positively ionized under an extremely high electric field, normally in the range of 10s V/nm. The bonds formed between adjacent atoms can be broken under the high field in conjunction with a short laser pulse to control the evaporation process to obtain a low detection rate. The detector cannot handle high detection rates, due to dead-time limitations. Usually, a successful laser pulse will dislodge one single fragment (ion). This is called a single event. In contrast, the process in which more than two ions are generated within one laser pulse is denoted

as “multiple events”. The ratio of the multiple events to the total events is called the “probability of multiple events (PME)”. We have measured more than 80 materials and found that all compounds which employ MVB show an abnormally high value of PME [Zhu et al., *Adv. Mater.* 2018, 30, 1706735]. Thus, the high PME value is utilized as an indicator of multivalent bonding.

Figure R1. PME value plotted on the basal plane of electrons transferred (ET) and electrons shared (ES). ET and ES are two natural variables describing the chemical bonding calculated by quantum chemical tools. More details about the ET-ES map can be found in Ref [Raty et al., *Adv. Mater.* 2018, 31, 1806280; Cheng et al., *Adv. Mater.* 2019, 31, 1904316].

In **Figure R1** and **Figure S3**, a map is shown, where the PME value is shown for different compounds depicted in a map that has been developed recently. This map separates different bonding mechanisms based on material properties including the electrical conductivity σ , the Born effective charge Z^* as a measure of the chemical bond polarizability, the effective coordination number, the optical dielectric constant ϵ_∞ as a measure of the electronic polarizability as well as the Grüneisen parameter as a measure of the bond anharmonicity. Different material classes have been classified based on unique sets of properties [Schön et al., *Sci. Adv.* 2022, 8, eade0828]. These materials can also be characterized by quantum-chemical bonding descriptors. As shown in Figure R1, all multivalent compounds possess high PME values. Hence, there is clearly a correlation between multivalent bonding and bond breaking in the atom probe. We note in passing an interesting detail that is relevant in this context. Amorphous chalcogenides such as amorphous GeTe do not show large PME values and have the characteristic properties of covalently bonded solids. This supports the idea that the bond rupture and the bonding mechanism are closely interwoven. Yet, you specifically ask for a potential causality and not just a (strong) correlation. We have not finalized our experiments yet, but can already offer

one preliminary answer. There is clearly a remarkable correlation between the electrical conductivity and the bond rupture as shown in **Figure R2**. This figure shows that only metavalent compounds with electrical conductivities around $10^3 - 10^4$ S/cm show high PME values. Doped semiconductors like P-doped Si, which have similar conductivities as well as low-conductive solids and good metals on the contrary do not show high PME values. Hence, showing high PME values is the exception and not the rule for solids. Interestingly, the range of electrical conductivities where the bond rupture is unusual is also the range where the transition between electron delocalization as in metals and electron localization as in covalent and ionic solids is found. This is in line with the notion that metavalent bonding is the competition zone between electron delocalization and electron localization. Yet, the detailed mechanisms underpinning the high PME are still elusive due to the very complex evaporation process involving high electrical fields, surface states, band bending, and the interaction between specimen and laser, etc. We have found some possible correlations between the high PME and the large penetration depth of the electric field, which depends on the dielectric constant and effective masses of materials according to the Debye screening length for semiconductors. Interestingly, MVB compounds are characterized by large dielectric constants and small effective masses, leading to a large penetration depth of the electric field. In addition, metavalent bonding is a kind of soft bond, which could be more unstable under high fields and laser excitations and, thus, several bonds break under one laser pulse generating multiple events. These plausible explanations need more rigorous verification and will not be discussed in this work.

Figure R2. Probability of Multiple Events (PME) determined by atom probe tomography plotted against the room temperature conductivity as $\log(\sigma)$. Unpublished work.

Revisions: We have added some of these thoughts in the Supplementary Information (Figure S3) and the revised manuscript on page 7.

3. Will the charge scattering at GBs in non-MVB materials differ from the MVB compounds? The collapse of metavalent bonding should only be valid in MVB compounds.

Response: We agree with the reviewer that the collapse of metavalent bonding should only be valid in MVB compounds. Yet, this does not weaken the generality and significance of this work. The number of compounds that have been identified to be metavalent is large and increasing. Such compounds include mono-chalcogenides, sesqui-chalcogenides, and even halide perovskites. These materials show promising applications in thermoelectrics, phase-change materials, topological insulators, and photovoltaics, etc. The findings about the charge carrier scattering at GBs in MVB materials are thus of great significance for a broad range of applications relevant to charge carrier transport. We also find the question of the reviewer interesting and insightful. This encourages us to extend our research objects to non-MVB compounds, which could enrich the mechanisms of charge scattering at GBs. We do not have experimental data for non-MVB materials at present and look forward to discussing such data in future publications.

Revisions: We highlighted the significance of this work on the charge carrier transport relevant applications of MVB compounds in the “Discussion” part on page 12.

Reviewer #3 (Remarks to the Author):

Recently, the effect of grain boundaries has attracted increasing attentions in thermoelectric community. A direct observation of how the grain boundary affect transport properties are challenging. In this work, a direct measurement on how a single grain boundary affect electrical transport property is realized. This is the highlight of this manuscript. The results are in detail and the discussion are reasonable. Therefore, I think this manuscript can be accepted for publication after addressing following questions.

General Response: We are grateful for the positive evaluation and recommendation of this work. We also thank the reviewer for the constructive comments. The responses to each question can be found below.

1. How to ensure the accuracy of determining GB potential barrier height? In Figure 2B, the three red data apparently show larger slope than three blue data. But for three red data, I think it is hard to tell who has larger negative slope, especially for $\theta = 41^\circ$ and $\theta = 54^\circ$. It seems that the result

largely depends on how the author fit the data. If so, the reported exact value of E_b might be misleading for readers. Any comment or supplement on this issue?

Response: Thanks for this helpful suggestion/question. The GB potential barrier height was determined by fitting the data points from low temperature to the inflection point where an inverse of the temperature dependence was observed. Since these data do not perfectly follow a linear behavior in Figure 2B, the final fitting slope (E_b value) indeed depends on how the data are fitted. We have modified the fitting code to allow the variation of the fitting confidence coefficient and the cut-off points. By doing so, we can obtain an error bar for each GB potential barrier height. The results are updated in Figure 4c. As commented by the reviewer, the differences in E_b for LAGBs and HAGBs are prominent. Yet, the values among the HAGBs show, within error bars, no obvious trends. This can be understood from the structural models of GBs. LAGBs can be described by dislocation models, where the dislocation density is inversely proportional to the GB misorientation angle. Thus, the GB energy, the density of trapping states, and the GB potential barrier height should increase with increasing misorientation angle for LAGBs. In contrast, HAGBs are more complex and cannot be described by dislocations or other structural models. The environment at the HAGBs varies with the misorientation angle and the GB plane indices without a clear trend besides some special GBs such as twin boundaries. This is in line with our findings.

Revisions: We added an error bar for each grain boundary potential barrier height in Figure 4c.

2. Will the grain boundary scattering be related to carrier concentration (Fermi level)? I notice that the samples in this work have carrier concentration much lower than the optimization for thermoelectric performance. Is this an intentional choose? Some explanation or discussion are suggested.

Response: According to the trapping state model (Equation 4), the GB potential barrier height is related to the carrier concentration, which is included in the parameter “N”. According to the definition, this N value is the concentration of ionized impurity atoms (usually the dopants). The concentration of dopants is the same for all the micro-scale specimens in this work because they were lifted out from the same bulk sample. APT results also show a similar composition of Ag in the matrix for both LAGB and HAGB samples. Assuming that the N value corresponds to the measured Hall carrier concentration (Figure S1), the conclusion is still valid. Thus, the impact of carrier concentration in this work can be excluded. Note that the obtained GB potential barrier height varies by changing the doping content. In this work, we choose Ag as the dopant because it has been widely used in thermoelectrics and it can often decorate GBs, which is important for the visualization of different GB features in the APT results. The content of Ag should be low (PbTe-0.005 Ag) to avoid the formation of complex structural defects and second-phase precipitates at GBs [Yu et al., *Nano Energy* 2022, 101, 107576].

Revisions: We explained the reason for choosing the desired composition in the “Methods” part on page 13.

3. The authors said MVB compounds show high dielectric constants. Is this a rigorous conclusion? Or, PbTe is just one of special cases? I think more discussion on this issue is necessary. This is critical for the main conclusion of the manuscript.

Response: Yes, this is a rigorous conclusion. MVB in chalcogenides like PbTe describes the σ -bond formed by a single p-electron (half an electron pair). Each atom in PbTe has octahedral coordination. The number of valence electrons in p-orbitals forming σ -bonds per atom is just three. Thus, for adjacent atoms only half of an electron pair, i.e. in total one electron is available to form a bond. This special bonding configuration gives rise to a unique portfolio of properties including high dielectric constants [Wuttig et al., *Adv. Mater.* 2018, 30, 1803777]. With the help of quantum mechanical calculation tools, chemical bonding can be described using two natural descriptors, i.e., the electrons transferred (ET) and electrons shared (ES) between the bonding atoms. **Figure R3** shows material properties plotted on the basal 2D plane of the ES-ET map. A large Born effective charge and a high optical dielectric constant are universal properties of MVB compounds. The dielectric constant defines the polarizability of electrons and ions in the presence of an electric field. This polarizability of electrons depends largely on the degree of delocalization of valence electrons. It is frequency dependent and shows a high-frequency component due to the electronic polarizability, summarized as the optical dielectric constant (ϵ_∞). The valence electrons in MVB compounds are rather delocalized due to the large overlap between p-electrons, which show strong polarizability and thus a large ϵ_∞ . MVB compounds also show a high Born effective charge that describes the chemical bond polarizability, leading to a large splitting between the longitudinal and transverse optical phonons. According to the Lyddane-Sachs-Teller-Relation (Equation 6), a large static dielectric constant is obtained as a consequence in MVB compounds.

Figure R3. 3D maps defining design rules for materials with desired properties. Figure taken from Ref. [Raty et al., *Adv. Mater.* 2018, 31, 1806280].

Revisions: We discussed in more detail the bonding origin of large dielectric constants in MVB compounds on pages 9 and 10.

4. Following question 3. I can understand the logic that a reduced dielectric screening ability leading to lower carrier mobility in GB due to MVB collapse. However, will this logic be contradicted to the case that lots of covalent compounds (such as Si and InSb) usually show high carrier mobility than MVB compounds?

Response: The high carrier mobility of different compounds stems from different aspects of mechanisms including the dielectric screening, the content of impurities, and the effective mass of charge carriers. The high carrier mobility in Si is often obtained in the ultra-pure single-crystal

Si. Introducing dopants such as P and B in Si will significantly decrease the carrier mobility, which is about one order of magnitude lower than the mobility of the Ag-doped PbTe at room temperature with a similar carrier concentration value, as shown in **Figure R4**. This is due to the strong ionized impurity scattering of dopants since the static dielectric constant of Si is quite low ($\epsilon_{st}=13$). In contrast, high carrier mobility is achieved in Ag-doped PbTe single crystal (Figure 2A). The ionized impurity scattering in general does not reduce the carrier mobility of PbTe significantly due to the strong dielectric screening effect ($\epsilon_{st}=380$). On the other hand, the high carrier mobility of InSb originates from its extremely small effective mass of electrons ($0.014 m_e$) due to its narrow bandgap. Note that the high carrier mobility in InSb is obtained at a very low doping level with a small charge carrier concentration (e.g. $3.5 \times 10^{15} \text{ cm}^{-3}$, [Ohshita, *Japanese Journal of Applied Physics* 1971, 10, 1365]). Thus, the conclusions in this work are not contradictory to other covalent compounds that show high carrier mobility.

Figure R4. Charge carrier mobility as a function of carrier concentration in (A) P-doped Si, and (B) B-doped Si. Figures are adapted from Ref [Masetti et al., *IEEE Transactions on electron devices* 1983, 30, 764-769].

Revisions: We added one example of doped Si on page 10 to demonstrate the strong ionized impurity scattering due to the weak dielectric screening effect.

5. The scope of current title is too large. I suggest the material PbTe to be clarified in the title.

Response: Thank you for this suggestion.

Revisions: We have modified the title to “Strong Charge Carrier Scattering at Grain Boundaries of PbTe caused by the Collapse of Metavalent Bonding”.

REVIEWERS' COMMENTS

Reviewer #1 (Remarks to the Author):

All the questions are answered. Now the manuscript can be accepted as it is.

Reviewer #2 (Remarks to the Author):

My comments have been addressed, I think it can be accepted.

Reviewer #3 (Remarks to the Author):

I have carefully read the revised manuscript and the rebuttal letter. Overall, the authors have properly addressed the raised issues by the reviewers, and the manuscript has been revised accordingly. Therefore, I think the current version of this manuscript can be accepted.

REVIEWER COMMENTS

Reviewer #1 (Remarks to the Author):

All the questions are answered. Now the manuscript can be accepted as it is.

Response: We sincerely appreciate the positive evaluation of our work. It is pleasing to read that our work can be published.

Reviewer #2 (Remarks to the Author):

My comments have been addressed, I think it can be accepted.

Response: We thank the reviewer for the support of the publication of this work.

Reviewer #3 (Remarks to the Author):

I have carefully read the revised manuscript and the rebuttal letter. Overall, the authors have properly addressed the raised issues by the reviewers, and the manuscript has been revised accordingly. Therefore, I think the current version of this manuscript can be accepted.

Response: We are grateful for the positive evaluation and recommendation of this work.